# Highly Efficient Transfer Hydrogenation of Biomass-Derived Furfural to Furfuryl Alcohol over Mesoporous Zr-Containing Hybrids with 5-Sulfosalicylic Acid as a Ligand

**DOI:** 10.3390/ijerph19159221

**Published:** 2022-07-28

**Authors:** Jirui Yang, Haixin Guo, Feng Shen

**Affiliations:** Agro-Environmental Protection Institute, Ministry of Agriculture and Rural Affairs, No. 31, Fukang Road, Nankai District, Tianjin 300191, China; yangjirui@caas.cn (J.Y.); haixin.guo.c8@tohoku.ac.jp (H.G.)

**Keywords:** biorefinery, lignocellulose, biomass-derived feedstock, organic ligand, Meerwein–Ponndorf–Verley

## Abstract

The catalytic transfer hydrogenation of biomass-derived furfural to furfuryl alcohol under mild conditions is an attractive topic in biorefinery. Herein, mesoporous Zr-containing hybrids (Zr-hybrids) with a high surface area (281.9–291.3 m^2^/g) and large pore volume (0.49–0.74 cm^3^/g) were prepared using the biomass-derived 5-sulfosalicylic acid as a ligand, and they were proven to be highly efficient for the Meerwein–Ponndorf–Verley reduction of furfural to furfuryl alcohol at 110 °C, with the highest furfuryl alcohol yield reaching up to 97.8%. Characterizations demonstrated that sulfonic and carboxyl groups in 5-sulfosalicylic acid molecules were coordinated with zirconium ions, making zirconium ions fully dispersed, thus leading to the formation of very fine zirconia particles with the diameter of <2 nm in mesoporous Zr-hybrids. The interaction between the 5-sulfosalicylic acid ligands and zirconium ions endowed mesoporous Zr-hybrids with relatively higher acid strength but lower base strength, which was beneficial for the selective reduction of furfural to furfuryl alcohol. A recycling study was performed over a certain mesoporous Zr-hybrid, namely meso-Zr-SA15, demonstrating that the yield and selectivity of furfuryl alcohol remained almost unchanged during the five consecutive reaction cycles. This study provides an optional method to prepare hybrid catalysts for biomass refining by using biomass-derived feedstock.

## 1. Introduction

The valorization of lignocellulosic biomass derived from agricultural and forestry residues into fine chemicals and fuels via catalytic conversion processes is an important issue in biorefinery amidst the growing concerns over energy security issues and carbon dioxide emissions driven by fossil fuels [1,2]. Furfural is a representative platform molecule commercially produced from the acid hydrolysis of pentosan sugars constituting the hemicellulose fraction of lignocellulose [3,4]. Furfuryl alcohol, accounting for around 65–75% of the global furfural consumption annually [5,6,7], is the most important furfural-derived product widely applied in the chemical industry [5]. Nowadays, furfuryl alcohol is industrially produced through the catalytic hydrogenation of furfural, which is performed over a copper chromite catalyst under high hydrogen pressure (>7 MPa) [8]. In light of the growing concerns over hydrogen safety and the toxicity of chromium species, a sustainable hydrogenation method is highly desired. Although the selective hydrogenation of furfural to furfuryl alcohol can be achieved with noble metal catalysts at ambient temperature under low hydrogen [9], the high cost and scarce reserves of noble metals greatly limit their practical application.

The Meerwein–Ponndorf–Verley (MPV) reduction has been widely reported to be an efficient transfer hydrogenation method for the selective reduction of carbonyl compounds at mild conditions, which is often performed over Lewis acidic or basic heterogenous catalysts using secondary alcohols as the reductant [10]. Zirconium (Zr)-containing materials are representative Lewis acid catalysts, mainly including ZrO_2_ [11], Zr-beta [12], and Zr-containing inorganic–organic hybrids (Zr-hybrids) [13,14,15,16,17,18]. Zr-hybrids are often prepared via precipitation and sol–gel methods based on the assembly of organic ligands with metal ions [16,19,20], and they have been widely reported to have a superior performance on the redox conversion of biofuranics compared to bare ZrO_2_ [14,15,16,17,18,19,21,22,23,24]. Generally, ligands with different molecular structures and coordinating groups result in a notably different catalytic performance in the MPV reduction of furfural [25]. The organic ligands can be derived from fossil fuel and biomass feedstocks. Considering the sustainability and cost advantages, biomass-derived molecules such as phytic acid [14], 2,5-furandicarboxylic acid [16,19], and tannin [18] seem to be more attractive compared to their fossil fuel-derived counterparts [17].

As a potential candidate for developing highly active Zr-hybrids, the biomass-derived 5-sulfosalicylic acid, containing three potential coordinating groups (i.e., –SO_3_H, –COOH, and –OH) in its molecule has been widely used as a ligand to stabilize metal ions [26,27,28]. Accordingly, 5-sulfosalicylic acid is expected to be closely coordinated with zirconium ions. Aside from the close interaction of organic ligands with zirconium ions, a high surface area and large pore volume are also needed to improve the exposure of active sites [29]. Moreover, mesopores in the hybrids are commonly thought to be favorable to mass transfer in the liquid phase reaction [30]. Herein, to efficiently catalyze the MPV reduction of furfural to furfuryl alcohol at mild conditions, mesoporous Zr-hybrids were prepared through the combination of the precipitation and soft templating method, in which 5-sulfosalicylic acid and the F127 triblock copolymer were employed as the ligand and pore-forming agent, respectively. The as-prepared mesoporous Zr-hybrids were proven to be very stable and highly efficient at 110 °C with the highest furfuryl alcohol yield reaching up to 97.8%. Extensive characterizations were performed to investigate the relationship between the physicochemical properties of the catalyst and its catalytic performance. The studies on the reaction parameter optimization were also performed. This study provides the optional method to prepare a hybrid catalyst for biomass refining by using biomass-derived feedstock.

## 2. Materials and Methods

### 2.1. Materials

ZrOCl_2_·8H_2_O (98%), 5-sulfosalicylic acid (99%), furfural (99%), and furfuryl alcohol (98%) were purchased from Aladdin Industrial Inc. (Shanghai, China). Nanoscale zirconium dioxide (nano-ZrO_2_, 99%), Pluronic F127 triblock copolymer (EO_106_PO_70_EO_106_, MW = 12,600), analytically pure 2-propanol, and ethanol were purchased from Macklin Co. Ltd. (Shanghai, China). Ultra-pure water (18.2 MΩ/cm) was produced with a Milli-Q purification system.

### 2.2. Synthesis of 5-Sulfosalicylic Acid-Derived Zr-Hybrids

In a typical procedure, 10 g ZrOCl_2_·8H_2_O was dissolved in 100 mL ethanol by stirring (1000 r/min) at room temperature for 8 h in a beaker. Afterward, 10 g of F127 was added into the solution and magnetically stirred (1000 r/min) for another 1 h until it was totally dissolved. A total of 7.5, 15, or 22.5 mL of 5-sulfosalicylic acid aqueous solution (7.4 g in 100 mL ultra-pure water) was added dropwise in the beaker to form a white paste within 15 min. Then, the beaker containing the paste was dried at 150 °C for 24 h in an oven. The dried solid sample was washed with boiling ethanol for 24 h to remove impurities and extract F127. Finally, the solid sample was obtained by filtration, dried at 60 °C for 12 h, and ground into a fine powder. Accordingly, the as-prepared mesoporous Zr-hybrids were denoted as meso-Zr-SA7.5, meso-Zr-SA15, and meso-Zr-SA22.5, respectively. For comparison, the sample denoted as Zr-SA15 was prepared with the addition of 15 mL of 5-sulfosalicylic acid aqueous solution, but without using F127.

### 2.3. Characterization

A scanning electron image (SEM) was recorded on a Hitachi S4800 instrument. A high-angle annular dark-field scanning transmission electron microscopy image (HADDF-STEM) was performed on a FEI Talos F200X instrument. X-ray diffraction (XRD) measurements were recorded on a Bruker D8 Advance diffractometer using Cu K*α* radiation. Fourier transform infrared spectroscopy (FTIR) measurements were conducted on a Bruker TENSOR 27 instrument in the range of 4000–400 cm^−1^. X-ray photoelectron spectroscopy (XPS) analysis was performed on an ESCALAB-250 X-ray photoelectron spectrometer. The N_2_ adsorption–desorption experiment was performed on a Micromeritics ASAP 2020 apparatus pyridine adsorption FTIR was performed on a Nicolet 380 apparatus. The sample was degassed under vacuum at 150 °C for 2 h, and then cooled to 25 °C. Pyridine adsorption was conducted at 25 °C followed by desorption at 150 °C. The Zr content in the sample was determined using inductively coupled plasma-optical emission spectroscopy (ICP-OES) on a PerkinElmer Optima 5300 DV instrument.

### 2.4. MPV Reduction of Furfural to Furfuryl Alcohol

Typically, 1 mmol furfural and 100 mg of catalyst were mixed with 10 mL of 2-propanol in a 30 mL Teflon-lined stainless steel autoclave and tightly sealed, followed by heating to a designated temperature (90–150 °C). After magnetically stirring (1000 r/min) for the desired time (1–5 h), the autoclave was cooled down to room temperature with tap-water. The slurry was filtered and the liquid products were quantitatively analyzed on a gas chromatograph (Shimadzu, GC-2010 plus) by using an external standard [31]. Each reaction was run in duplicate, and average values were reported with a relative deviation below 5%. The calculation of furfural conversion, furfuryl alcohol yield, and selectivity was carried out according to Equations (1)–(3). GC-MASS (Agilent 7890B-5975C) was used to identify the intermediates in the reaction process. To evaluate the catalyst recyclability, the spent catalyst was recovered by centrifugation, washed with 20 mL ethanol twice, and employed for the next run.
(1)$$\mathrm{Furfural}\;\mathrm{conversion}\;(\%)=(1-\frac{\mathrm{moles}\;\mathrm{of}\;\mathrm{furfural}\;\mathrm{in}\;\mathrm{products}}{\mathrm{moles}\;\mathrm{of}\;\mathrm{loading}\;\mathrm{furfural}})\;\times \;100\%$$
(2)$$\mathrm{Furfuryl}\;\mathrm{alcohol}\;\mathrm{yield}\;(\%)=\frac{\mathrm{moles}\;\mathrm{of}\;\mathrm{furfuryl}\;\mathrm{alcohol}\;\mathrm{produced}}{\mathrm{moles}\;\mathrm{of}\;\mathrm{loading}\;\mathrm{furfural}}\times \;100\%$$
(3)$$\mathrm{Furfuryl}\;\mathrm{alcohol}\;\mathrm{selectivity}\;(\%)=\frac{\mathrm{furfuryl}\;\mathrm{alcohol}\;\mathrm{yield}\;(\%)}{\mathrm{furfural}\;\mathrm{conversion}\;(\%)}\times \;100\%$$

## 3. Results and Discussion

### 3.1. Characterization Results

The FTIR spectra were first analyzed to describe the interaction between the components (Figure 1). For pure 5-sulfosalicylic acid, the band at 1473 cm^−1^ corresponded to the skeletal vibration of the benzene ring bonds [27]. The band at 655 cm^−1^ arises from the C-S bond vibration of the sulfonic group [17]. The band at 1089 cm^−1^ corresponded to the symmetric stretching vibration of the C–C bond between the aromatic ring [32]. These bands above can all be observed in the mesoporous Zr-hybrids, indicating that the molecular structure of 5-sulfosalicylic acid is not significantly affected in the preparation process. The band at 1677 cm^−1^ in 5-sulfosalicylic acid was assigned to the stretching band of the carboxyl group and the band at around 3380 cm^−1^ was the hydroxyl group [26]. The bands at 1161 cm^−1^ and 1033 cm^−1^ were assigned to the symmetric stretching vibrations of the sulfonic group in 5-sulfosalicylic acid [27]. However, the FTIR spectra of meso-Zr-SA7.5 revealed that the characteristic bands for sulfonic (1161 and 1033 cm^−1^) and carboxyl (1677 cm^−1^) groups almost disappeared, suggesting that both the sulfonic and carboxyl groups were coordinated with Zr^4+^ [26]. The intensities of the bands corresponding to sulfonic group (1161 and 1033 cm^−1^) became stronger with the increasing dosage of 5-sulfosalicylic acid in the preparation of meso-Zr-SA15 and meso-Zr-SA22.5. Moreover, a broad peak at 520–450 cm^−1^ in the mesoporous Zr-hybrids could be assigned to the Zr–O bond vibration showing the existence of amorphous zirconia [33]. The preparation of mesoporous Zr-hybrids started from the ethanol solution containing the dissolved zirconium alkoxide and F127 micelles. After the addition of the 5-sulfosalicylic acid solution (pH ≈ 1.3), both the hydrolysis of zirconium alkoxide and the subsequent condensation process sped up under the acidic condition [34]. Meanwhile, the hydrolyzed hydrophilic inorganic zirconium species interacted with the F127 micelles via hydrogen bonding and weak coordination bonds [35] and were coordinated with the 5-sulfosalicylic acid molecules. Finally, the gelatinous precipitate was formed as the hybrid precursor. Here, the F127 copolymers acted as the organic template and were extracted with boiling ethanol after the hybrid precursor was dried, thus giving rise to the formation of mesoporous Zr-containing hybrids.

The elemental mappings from the SEM image were performed to investigate the distribution of elements on the surface of mesoporous Zr-hybrids. The images (Figure 2a–e) for the meso-Zr-SA15 revealed that all elements including C, O, S, and Zr were evenly dispersed on the hybrid, most likely due to the assembly of 5-sulfosalicylic acid and Zr^4+^ in the initial preparation step [18]. No obvious zirconia nanoparticles could be observed from the HADDF-STEM image for meso-Zr-SA15 (Figure 2f), indicating the formation of very fine zirconia particles with a diameter of <2 nm since Zr^4+^ is extremely evenly dispersed by 5-sulfosalicylic acid. The wide-angle XRD pattern shows that meso-Zr-SA15 is amorphous with only a few broad diffractions (Figure 2g), although there is an intense peak at 7.5° corresponding to the (111) reflection of UiO-66(Zr) [16,36]. The small-angle XRD pattern showed that there were no obvious diffraction peaks in the meso-Zr-SA15 (Figure 2h), suggesting that there is no ordered porous structure [37]. Other Zr-hybrids including meso-Zr-SA7.5, meso-Zr-SA22.5, and Zr-SA15 displayed similar results for the SEM, HADDF-STEM, and XRD measurements.

The nitrogen adsorption–desorption isotherms of meso-Zr-SA7.5, meso-Zr-SA15, and meso-Zr-SA22.5 showed IV-type curves with sharp capillary condensation steps in the relative pressure range of 0.65–0.95 and H_2_-type hysteresis loops (Figure 3), indicating the typical 2D hexagonal mesoporous structures [35]. The BET surface area, pore volume, and pore width were calculated and summarized in Table 1. With the increasing amount of 5-sulfosalicylic acid solution from 7.5 to 22.5 mL in the initial preparation step, the BET surface area and pore volume of the mesoporous Zr-hybrids were first increased and then decreased, with the maximum values reaching 319.5 m^2^/g and 0.74 cm^3^/g in meso-Zr-SA15, respectively (Table 1). The result suggested that the excess 5-sulfosalicylic acid might occupy some of the pores and lead to a slight reduction in the surface area. The pore width of the mesoporous Zr-hybrids was in the range of 6.4–10.9 nm. In comparison, Zr-SA15 prepared in the absence of F127 had a low BET surface area (28.1 m^2^/g) and ignorable pore volume (0.02 cm^3^/g). Therefore, using F127 as the pore-forming agent to introduce mesopores can greatly enhance the BET surface area for Zr-hybrids, thus providing more active sites for the substrates [38]. The BET surface area of the commercial nano-ZrO_2_ was measured to be 20.6 m^2^/g (Appendix A).

The XPS analysis was conducted to compare the Lewis acid strength of Zr in Zr-hybrids with that in the bare nano-ZrO_2_ (Figure 4). For the bare nano-ZrO_2_, two peaks at 182.2 and 184.6 eV corresponded to Zr 3d_5/2_ and 3d_3/2_, respectively. These two peaks shifted to higher levels at 182.8 and 185.2 eV for the Zr-hybrids, meaning that Zr atoms in the Zr-hybrids had a higher positive charge, thus manifesting a stronger Lewis acidity, which can be attributed to the interaction of organic ligands and metal ions [14,15,18]. Furthermore, the Zr-hybrids showed higher O1s binding energy compared to the nano-ZrO_2_ (531.6 vs. 530.1 eV), suggesting that O atoms in the Zr-hybrids had lower negative charge corresponding to the lower basicity [19].

The pyridine-adsorbed FTIR spectra were performed at the desorption temperature of 150 °C to determine the Lewis and Brønsted acid sites in the samples (Figure 5). For nano-ZrO_2_, the peaks at 1452 and 1610 cm^−1^ corresponded to the pyridine interacting with the Lewis acid [18]. These peaks shifted to lower values of 1448 and 1607 cm^−1^ in the Zr-hybrids due to the pyridine interacting with coordinated Zr [16]. The peaks at 1540 and 1490 cm^−1^ were attributed to the pyridine interacting with the Brønsted acid and both of these two acid sites, respectively [18]. The quantitative analysis results showed that mesoporous Zr-hybrids including meso-Zr-SA7.5, meso-Zr-SA15, and meso-Zr-SA22.5 had a comparable amount of the Zr element in the samples (35.5–37.5%) and contained Lewis acid sites ranging from 47.9–53.3 μmol/g (Table 1). In contrast, the number of Lewis acid sites in nano-ZrO_2_ reached up to 71.3 μmol/g, probably due to the higher Zr content (74.0%) (Appendix A). The hybrid Zr-SA15 had relatively lower Lewis acid sites compared with the meso-Zr-SA15 (44.9 vs. 52.6 μmol/g). This result might be attributed to the much smaller surface area (28.1 m^2^/g) and pore volume (0.02 cm^3^/g) in Zr-SA15 (Table 1), thus reducing the accessible Zr site to some extent. The number of Brønsted acid sites in all samples was in the range of 5.6 to 9.5 μmol/g (Table 1 and Appendix A).

### 3.2. MPV Reduction of Furfural to Furfuryl Alcohol

First, meso-Zr-SA15 was employed to study the impact of temperature on the MPV reduction of furfural within 1 h (Figure 6). With the temperature increasing from 90 to 150 °C, the yield and selectivity for furfuryl alcohol were increased from 16.1% to 92.5% and 24.6% to 93.9%, respectively, suggesting that a higher temperature helped to accelerate the transformation of furfural into furfuryl alcohol [21]. Meanwhile, it was worth noting that a furfuryl alcohol selectivity of 55.9% could be achieved with a high furfural conversion of 77.4% at 110 °C. Next, the catalytic performance of meso-Zr-SA15 was investigated at 110 °C for a longer time with the results shown in Figure 7a. It was observed that furfural conversion was increased with the reaction time extended from 1–5 h. The furfuryl alcohol yield and selectivity were first increased with the reaction time from 1 to 4 h, and then decreased at 5 h, giving the highest furfuryl alcohol yield of 97.6% with the selectivity of 98.7% at 4 h. At higher temperatures of 130 °C and 150 °C, the reduction in furfuryl alcohol yield and selectivity was observed at 3 h and 2 h, respectively (Appendix A). Based on the results, it was indicated that the maximum furfuryl alcohol yield was obtained at a mild temperature of 110 °C, and a long reaction time and high temperature easily led to the reduction in furfuryl alcohol selectivity, which was mainly attributed to the occurrence of side reactions (details below).

The performance of mesoporous Zr-hybrids prepared with different 5-sulfosalicylic acid dosages was compared at 110 °C. The compared results showed that meso-Zr-SA15 exhibited the best catalytic performance, and based on that, a furfural conversion of 98.8% and furfuryl alcohol yield of 97.6% were achieved after reaction for 4 h. In the case of meso-Zr-SA7.5, the highest furfural conversion (96.3%) and furfuryl alcohol yield (93.5%) could also be obtained at 4 h (Figure 7b). For meso-Zr-SA22.5, the highest furfuryl alcohol yield of 88.9% and selectivity of 92.2% were obtained at 3 h, which were much lower compared with those of meso-Zr-SA7.5 and meso-Zr-SA15 (Figure 7c). To further investigate the differences in catalytic performance, the by-products were determined by GC-MASS after reaction over mesoporous Zr-hybrids for 4 h (Appendix A). One peak at a retention time of 3.881 min was inferred as 2-(isopropoxy)methyl furan, which resulted from the reaction of furfuryl alcohol with the abundant 2-propanol catalyzed by the Lewis or Brønsted acid [16,39]. The other peak at a retention time of 5.434 min was identified as isopropyl levulinate through a comparison against the standard compound, which arose from the alcoholysis of 2-(isopropoxy)methyl furan catalyzed by the Brønsted acid [16,39]. Accordingly, a plausible reaction pathway in the MPV reduction of furfural over mesoporous Zr-hybrids was proposed, as shown in Figure 1. It was observed that the amounts of 2-(isopropoxy)methyl furan and isopropyl levulinate were increased with the increasing sum of Lewis acid and Brønsted acid from 54.5 to 62.8 μmol/g in the mesoporous Zr-hybrids (Table 1), according to the peak area of each chemical (Appendix A). Moreover, the meso-Zr-SA22.5 prepared with the highest dosage of 5-sulfosalicylic acid afforded the highest number of Brønsted acid sites (9.5 μmol/g, Table 1) for the further transformation of 2-(isopropoxy)methyl furan to isopropyl levulinate. Therefore, the reduction in furfuryl alcohol yield and selectivity over Zr-SA22.5 was mainly attributed to the transformation of furfuryl alcohol to more by-products.

When the dosage of meso-Zr-SA15 was increased to 200 mg, the furfuryl alcohol yield reached up to 97.8% within only 2 h at 110 °C (Table 2, entry 1–3), suggesting that meso-Zr-SA15 was highly efficient for the MPV reduction of furfural to furfuryl alcohol. In contrast, nano-ZrO_2_ afforded much lower furfural conversion (27.2%) and furfuryl alcohol yield (4.9%) at identical conditions (Table 2, entry 4), although nano-ZrO_2_ contained a higher number of Lewis acid sites compared with that in meso-Zr-SA15 (71.3 vs. 52.6 μmol/g). For the zirconia catalyst, the Zr^4+^-O^2−^ acid–basic pair played the most crucial role for catalyzing the MPV reduction [11]. The direct hydrogen transfer mechanism was proven to be the most favorable pathway in MPV reduction, which involved a concerted six-membered ring transition state formed from the coordination of both the H-acceptor and H-donor to a Lewis acid metal ion [40,41]. The stronger Zr^4+^ Lewis acidity means a larger electron affinity for the active Zr sites, causing stronger binding to the reactants and translating to a lower apparent activation energy, thus endowing the catalyst with higher activity for the MPV reaction [14,42,43]. The basic site acted on the deprotonation of alcohol molecules to assist in the formation of alkoxide species with Lewis acid sites [44]. The low-to-moderate basic strength was beneficial to the selective production of furfuryl alcohol from furfural by inhibiting side reactions such as condensation, especially catalyzed by strong basic species [16,19]. Based on the XPS spectra of Zr 3d and O 1s (Figure 4), meso-Zr-SA15 showed a relatively higher acid strength but lower base strength compared with the bare nano-ZrO_2_, suggesting that the former catalyst was more favorable for the selective MPV reduction of furfural to furfuryl alcohol [19]. Although the Brønsted acid site was reported to play a slightly positive role in MPV reduction [17,19], the presence of Brønsted acid sites was conducive to the occurrence of side reactions, resulting in the reduction of furfuryl alcohol yield, as above-mentioned.

A very low furfuryl alcohol yield of 1.1% with a furfural conversion of 35.9% was obtained over Zr-SA15 prepared without the addition of F127 (Table 2, entry 5). The poor catalytic performance of Zr-SA15 might be attributed to its much lower surface area (28.1 m^2^/g) and pore volume (0.02 cm^3^/g) (Table 1) because the number of surface metal ions available would be greatly reduced, along with the decline in surface area [11,13]. Therefore, a higher surface area and larger pore volume were two favorable factors to enhance the catalytic performance for meso-Zr-SA15. Moreover, 5-sulfosalicylic acid worked as the stabilizing agent, leading to the generation of very fine zirconia particles with the diameter of <2 nm in meso-Zr-SA15 (Figure 2f). Generally, heterogeneous catalysis occurred on the surface of metal particles, and the decrease in the size of metal particles resulted in a higher fraction of surficial atoms available, thus bringing about improvement on the catalytic performance [45]. The blank control experiment showed that 20.1% of furfural was converted with no furfuryl alcohol yield (Table 2, entry 6). It is generally accepted that the reversible acetalization of furfural with 2-propanol prevailed in the absence of a catalyst or over a Lewis acid catalyst with low activity [17]. As shown in Figure 6, furfuryl alcohol selectivity was increased within 1 h as the reaction temperature was increased from 90–150 °C, because the MPV reduction of furfural prevailed over the furfural acetalization reaction in the presence of an active meso-Zr-SA15 catalyst at higher temperatures. In conclusion, the outstanding catalytic performance of meso-Zr-SA15 was determined by multiple factors including the stronger Lewis acidity, the weaker basic strength, the proper content of the Brønsted acid, the higher surface area, and the larger pore volume. The catalytic performance of meso-Zr-SA15 in this study was compared with some remarkable catalysts used in selective hydrogenation of furfural to furfuryl alcohol (Table 3). It was found that an equally excellent catalytic performance was achieved over meso-Zr-SA15 at a relatively lower temperature of 110 °C, confirming that meso-Zr-SA15 is highly active and selective for the reduction of furfural to furfuryl alcohol.

### 3.3. Catalyst Recyclability

The hybrid meso-Zr-SA15 was chosen to further evaluate the catalyst recyclability. The result (Figure 8) showed that furfural conversion (74.8–75.4%), furfuryl alcohol yield (55.1–55.9%), and selectivity (73.7–74.4%) were almost unchanged in five consecutive cycles, suggesting that meso-Zr-SA15 had a good recyclability. The content of Zr in the filtrate of each run was detected to be below the detected limit (0.1 mg/L) by ICP-OES, indicating that the leaching of Zr was negligible due to the strong interaction of 5-sulfosalicylic acid and Zr^4+^ [50,51]. Moreover, the reaction was performed under a relatively low temperature (110 °C) so that the formation of humins could be greatly suppressed [52,53], avoiding deactivation of the catalyst accordingly. Similar results were also obtained when the reaction time was extended to 2 h (Appendix A).

## 4. Conclusions

In this study, highly efficient 5-sulfosalicylic-derived mesoporous Zr-containing hybrids were prepared for catalyzing MPV reduction of furfural to furfuryl alcohol at a mild temperature of 110 °C. Specifically, the maximum furfuryl alcohol reaching up to 97.8% could be achieved over meso-Zr-SA15 with the best activity, which was also tested to have a good recyclability. The outstanding performance of meso-Zr-SA15 was determined by multiple factors including the stronger Lewis acidity, the weaker basic strength, the proper content of Brønsted acid, the higher surface area and the larger pore volume. All these advantages mentioned above can be attributed to interaction between 5-sulfosalicylic acid ligands and zirconium ions and mesoporous structure produced by pore-forming agent. This study provides an optional method to prepare hybrid catalyst for biomass refining by using biomass-derived feedstock.

## Data Availability

The data presented in this study are available on request from the corresponding author.

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
