# Peer review of "Highly Efficient Transfer Hydrogenation of Biomass-Derived Furfural to Furfuryl Alcohol over Mesoporous Zr-Containing Hybrids with 5-Sulfosalicylic Acid as a Ligand"

_ijerph, 2022, doi:10.3390/ijerph19159221_

Round 1
Reviewer 1 Report
The work presented by Yang et al. is interesting and deals with a theme of great actuality and relevance. The work is well written and articulated but it requires some modifications before being accepted:
1) "The biomass-derived molecules such as phytic acid [14], 2,5-furandicarboxylic acid [16,19] and tannin [18] seem to be more attractive due to the sustainability and cost advantages compared with the fossil fuel-derived counterparts [17]." The authors should bettere contextualize this sentence in the Introduction.
2) The authors found that the addition of F127 strongly improved the textural properties of the catalyst. Why? How the F127 acted?
3) Runs 1-5, 6-10 and 11-14 in Table 2 should be reported in three different graphs in order to better show the kinetic of the reaction.
4) The authors have normalized the peaks deriving from the GC-MS analysis? It is not significant compare the peak area of the compounds in different runs.
Author Response
thanks

Reviewer 2 Report
Catalytic conversion of biomass derived furans or furan derivatives has been an active research area because of their relevance to different industrial sectors as platform chemicals. Here, Yang et al. demonstrates an effective conversion of furfural to furfuryl alcohol using hybrid zirconium catalyst containing 5-sulfosalicylic acid as a ligand. The authors insist that the hybrid catalyst facilitate mild catalytic conversion of furfural with high selectivity and the results could be meaningful, especially an interest in sustainable industrial biorefineries has been rapidly increasing. A few suggestions for the authors are listed below:
1. Introduction: The authors should be careful about the overall justification of the study. In particular, lignocellulosic biomass itself could not be carbon neutral – it would be better to emphasize the aspect of waste valorization (i.e., valorizing waste materials generated during the processing of lignocellulosic biomass via the proposed catalytic conversion).
2. Figs 1, 5: Please clearly denote functional groups corresponding to each wavenumber.
3. Fig 2: Please improve the resolution of a-e. If the goal of these subfigures are showing the high dispersion of functional groups within the catalyst, please provide a merged image as well.
4. Fig 6 and related discussion: The furfuryl alcohol selectivity increased as the reaction temperature was increased. However, the authors claim that long reaction time and high temperature lead to the reduction of furfuryl alcohol selectivity. Similarly, Table 2 indicates that an increase in reaction time could lead to an increased furfuryl alcohol selectivity at least to a certain degree.
5. Scheme 1: Please add furfural acetalization reaction in the scheme, which seemed to occur in negative control (nano-ZrO2).
6. Scheme 1: Also, somewhere in the scheme, please denote the influence of basic strength on the production of by-products.
7. Table 3: Please add more discussion concerning Ref. 14 and 18, which also reported a high furfural alcohol yield under a mild reaction condition. Isn’t the use of phytic acid or tannin in the catalyst fabrication more justifiable in terms of environmental sustainability?
8. Fig 7: The reusability tested was performed under mild conditions (110 degree C, 1 hr) showing a low furfural conversion rate (lower than 80%) and a furfuryl alcohol selectivity (~60%). What will the reusability of the hybrid catalyst look like when each reaction time is increased to 2 hr?
Author Response
Thank you very much for your decision and the comments. These comments are all valuable and very helpful for revising and improving our manuscript. We have revised the manuscript carefully and accordingly.

Round 2
Reviewer 1 Report
The manuscript has been improved and it is appropriate for publication.